# Collaborative innovation evolution of the logistics and manufacturing industry in China

Xiuling Chen[1,2], Haohua Liu[1]*

**1** School of Business Administration, Jiangxi University of Finance and Economics, Nanchang, Jiangxi, China, **2** Business School, Wuyi University, Nanping, Fujian, China

* 125054960@qq.com

**Data Availability Statement:** TThe collaborative innovation data used in this paper are collected and sorted by the authors themselves, and the method is as described in Section 3.1 of the paper. The data has been uploaded as supporting information

## Abstract

The logistics industry and the manufacturing industry are mutually productive factors and service objects, which determines that the two must develop cooperatively. In the increasingly fierce market competition, open collaborative innovation is more conducive to improving the level of linkage between the logistics industry and the manufacturing industry, and promoting industrial development. Based on the patent data of 284 prefecture-level cities in China from 2006 to 2020, this paper uses Gis spatial analysis, spatial Dubin model and other methods to analyze the collaborative innovation between the logistics industry and manufacturing industry. The results lead to several conclusions. (i) The overall collaborative innovation level is not high, and in terms of the evolution cycle, collaborative innovation has experienced three stages: embryo, rapid development and stable development. (ii) The spatial agglomeration characteristics of the collaborative innovation between the two industries are more and more obvious, and the Yangtze River Delta urban agglomeration and the middle reaches of the Yangtze River urban agglomeration play an important role in the collaborative innovation between the two industries. (iii) The hot spots of collaborative innovation between the two industries are concentrated in the eastern and northern coastal areas, while the cold spots are concentrated in the south of the northwest and southwest areas in the late stage of the study. (iiii) The positive influencing factors of local collaborative innovation between the two industries include economic development, scientific and technological level, government behavior, and employment, while the negative influencing factors include information technology level, and logistics infrastructure. Economic development has a negative spatial spillover effect on surrounding areas, while the spatial spillover effect of scientific and technological level is significantly positive. This article aims to explore the current situation and influencing factors of collaborative innovation between the two industries, with a view to proposing countermeasures and suggestions for improving the level of collaborative innovation between the two industries, and also providing new ideas for research on cross industry collaborative innovation.

S1.The patent data used in this article is from the CNKI patent database(https://kns.cnki.net/kns8?dbcode=SCOD).The data of the impact variables is sourced from the ESPDATA database (https://www.epsnet.com.cn).

**Funding:** This study was supported by Humanities and Social Science Research Project of Universities in Jiangxi Province (Grant No. GL20112), The 14th Five-Year Social Science Fund Project of Jiangxi Province (Grant No. 21YJ22D), and Social Science Fund of Fujian Province (Grant No. FJ2021X017). The funders had role in decision to publish the manuscript. Roles: Supervision, Project administration, Funding acquisition.

**Competing interests:** The authors have declared that no competing interests exist.

# 1. Introduction

In the process of implementing the innovation driven strategy and promoting the construction of an innovative country in China, how to effectively organize and coordinate collaborative innovation among industries to improve innovation performance is a topic worthy of attention. Manufacturing industry is an important supporting industry in China, but relying solely on production and processing can no longer maintain its competitive advantage. In the context of high-quality economic development, in order to accelerate the transformation of the manufacturing industry, it is necessary to improve its innovation ability and enhance its position in the global value chain. In the increasingly competitive global market, it is difficult for the manufacturing industry to rely on independent innovation to gain advantages. Open cooperative innovation with upstream and downstream industries in the supply chain is more conducive to improving industry performance [1]. As a productive service industry, logistics industry is closely related to the manufacturing industry. The essence of the coordinated development of the manufacturing and logistics industries is to work together to jointly operate and solve the problems of industrial development and upgrading [2]. Collaborative innovation between the manufacturing and logistics industries will effectively break the information asymmetry barriers between suppliers, producers, and sellers, and integrate their respective resources through various knowledge inputs, thereby creating a supply chain value superior to that of a single enterprise [3].

The Chinese government attaches great importance to the industrial integration and development of the logistics and manufacturing industries (hereinafter referred to as the two industries). As early as 2009, the State Council raised the linkage development project between the two industries to the national strategic level, and proposed many policies to support the industrial integration between the two industries. In 2020, 13 departments including the National Development and Reform Commission jointly issued the "Implementation Plan for Promoting the Deep Integration and Innovation Development of the Logistics and Manufacturing Industries", which mentions the need to actively promote the integration and innovation between the logistics and manufacturing industries, and promote synergy and linkage between the two. Collaborative innovation is one of the forms of industrial convergence between the two industries [4]. There are certain differences and complementarities in technology and knowledge between the two industries [3]. As a service industry, logistics mainly focuses on tacit knowledge such as information and management. Although tacit knowledge has a high degree of commonality, it is difficult to transmit and accumulate. The manufacturing industry is dominated by explicit knowledge such as engineering and applied technology, which is highly specialized but can be quickly accumulated and transferred [5]. Both the logistics industry and the manufacturing industry are at a time of transformation towards high-quality development. Collaborative innovation is the innovative cooperation between the two in technology, products, and organization [6], which can greatly improve the production efficiency and service quality of the industry. In the process of service-oriented transformation, the manufacturing industry forms a collaborative innovation relationship with the logistics industry through activities such as product innovation and service innovation, which helps to break the barriers between the two industries and promote the generation of new value in the supply chain. This has important significance for the transformation and upgrading of the manufacturing industry and the high-quality development of the logistics industry.

The following contents of this paper are arranged as follows: The first part introduces the literature review; The second part introduces the data source and method of this paper. The third part analyzes the evolution characteristics of collaborative innovation between the two industries; In the fourth part, the spatial Dubin model is used to analyze the influencing factors

of collaborative innovation between the two industries. The fifth part draws conclusions and puts forward countermeasures and suggestions.

## 2. Literature review

Collaborative innovation is a new paradigm for technological innovation, emphasizing the integration of various innovation elements and the unobstructed flow of innovation resources, thereby creating synergies that cannot be achieved by a single entity [7, 8]. Collaborative innovation can help enterprises make use of internal and external expertise, increase the amount and type of their innovation activities, and expand the effect of innovation activities [1, 9]. Traditional closed innovation has been unable to adapt to the increasingly competitive market environment [10], and the focus of enterprise innovation is gradually moving towards collaborative innovation [11]. Entities participating in collaborative innovation can not only reduce the costs and risks of independent R&D innovation, but also improve the performance of innovation [1]. Collaborative innovation is one of the important ways for enterprises to implement innovation strategies [9, 11].

In previous studies, collaborative innovation refers to two or more entities sharing knowledge and conducting research and development activities together during the innovation process [12, 13], so cooperative patents of multiple subject are mainly used as the measurement indicator of collaborative innovation [14, 15]. The object of this study is the logistics industry and manufacturing industry. The logistics industry itself is an important link in the manufacturing industry chain. Collaborative innovation between the two industries is essentially industry chain innovation, emphasizing interactive innovation between upstream and downstream nodes of the manufacturing industry chain [16]. In China, the proportion of logistics outsourcing in industrial, wholesale and retail enterprises has reached 69.7% in 2020, which indicates that a considerable part of manufacturing enterprises have not outsourced their logistics business and belong to self-operated logistics. The activities of self-operated logistics of these manufacturing enterprises belong to the logistics industry functionally, but they are dominated by manufacturing enterprises. The innovation activities and achievements of these manufacturing enterprises in self-operated logistics are still registered in the name of the manufacturing enterprises when the patent application is made, but these innovation achievements belong to the logistics industry in terms of content. Considering the remaining nearly 30% of self-operated logistics, the collaborative innovation studied in this article includes not only the innovative activities of logistics enterprises and manufacturing enterprises as two independent individuals that cooperate with each other, but also the innovative activities of manufacturing enterprises in the logistics field alone, and the innovative activities of logistics enterprises in the manufacturing field alone. This may more comprehensively depict the situation of collaborative innovation between logistics and manufacturing industries in China.

Regarding the influencing factors of collaborative innovation, relevant research involves the types of partners [9], technological proximity [17], enterprise innovation capability [18], stable relationships [19], government funding [20, 21], collaborative willingness [22], Economic development level [23], R&D investment [24], transportation infrastructure [25], etc., it can be seen that the research results in this part are relatively rich. This article uses "collaborative innovation" or "cooperative innovation" as the keyword or title of the article on CNKI, and searches for documents containing "logistics" and "manufacturing" in the keyword or title; In WOS, this article uses "collaborative innovation" or "innovation network" as the title for retrieval, and the keywords or title include "logistic" and "manufacturing". Only one relevant paper can be found through the two searches, which shows that there is very little literature on

collaborative innovation between the two industries. Collaborative innovation is one of the forms of integration of logistics and manufacturing industries. Therefore, regarding the research on the influencing factors of collaborative innovation between the logistics industry and the manufacturing industry, this article focuses on the influencing factors of the integration of the two industries. The refinement of social division of labor and the trade-off of transaction costs are considered to be the internal driving forces for the integration of the two industries [26]. In recent years, the factors mentioned in research that affect the integration between the two industries include technological innovation [27], integration ability [28], manufacturing value chain climb [29], tax system [30], technological research and development difference [31], enterprise research and development ability [32], contract environment [33], and so on. However, the research results in this field have not yet reached consistent conclusions. The possible reason is that the academic community has not yet formed a unified view on the measurement of collaborative innovation. Joint patents are a common measurement indicator, but the National Bureau of Statistics does not separately count joint patents. Existing literature using joint patents is searched, downloaded, and screened by the author before use. There are also several sources of data: patent retrieval and analysis platform of the China National Intellectual Property Administration of the People's Republic of China [34], CNKI patent database [16], Derwent World Patent Index [35], etc. These may lead to different joint patent data used by different authors even for the same research content. The issue of collaborative innovation between two industries studied in this article belongs to cross industry collaborative innovation. Currently, cross industry collaborative innovation is mostly studied using the evolutionary game method [36]. There is no industry data in the patent data released by China's national or regional statistics bureaus. The China National Intellectual Property Administration has provided patent databases for individual key industries, and patent data for cross industry cooperation need to be downloaded and sorted by authors, which requires a lot of work. In the 2017 national economy industry classification, the manufacturing industry includes 31 major categories, involving many industries. In the national key industry patent information service platform, only individual industries under the manufacturing industry such as equipment manufacturing industry and textile industry have been subject to patent statistics, without statistics on the patents of the entire manufacturing industry. Therefore, how to use joint patents as a measurement indicator of collaborative innovation between logistics and manufacturing industries, and how to conduct data statistics and screening are issues that need to be addressed in this article.

In the study of influencing factors of collaborative innovation, traditional quantitative analysis methods are commonly used, including the least square method [8], the Tobit model [37], and the negative binomial regression model [38, 39]. Other methods such as structural equation modeling [40, 41], and social network analysis [42] are also involved. There are few literatures using spatial econometric models. Spatial econometrics is a popular method in regional economics, economic geography, and other fields in recent years, which can effectively evaluate the spatial effects of economic activities. Regarding the impact of spatial factors on collaborative innovation, many articles have included geographical distance as an influencing factor in traditional regression models [43, 44], but this treatment can only assess the impact of spatial factors on collaborative innovation, and cannot assess the spatial effects of other influencing factors. This article uses the spatial econometric method to analyze the influencing factors of collaborative innovation, which can not only evaluate the impact of local influencing factors on local collaborative innovation between the two industries, but also evaluate the impact of local influencing factors on collaborative innovation between the two industries in surrounding areas.

Compared with previous studies, the possible marginal contributions are as follows: First, innovation in collaborative innovation measurement. Previous studies mostly used the number of multiple subject joint patent applications in a single industry as a measure of collaborative innovation. This paper takes patents filed by the two industries in each other's fields, which not only include the joint invention patents of the two industries but also better reflect the comparable proportion of logistics outsourcing and self-support in the Chinese manufacturing industry. It gives a new approach for the construction of a measurement index of collaborative innovation between two industries. The second is new applications of research methods. Compared with the singularity of the game model, this paper uses GIS space analysis and the spatial Durbin model to enrich the research methods of collaborative innovation. Third, prefecture-level cities are the research scale. Due to data problems, there are few studies using the prefecture-level city scale. In this paper, the zip codes in the patent data are matched to the prefecture-level cities, and thus the panel data of 284 prefecture-level cities are constructed for research. In conclusion, this paper has greatly promoted the theoretical research of industrial collaborative innovation and enriched its research methods.

## 3. Data sources and methodology

### 3.1 Data sources

The patent data used in this article is from the CNKI patent database(https://kns.cnki.net/kns8?dbcode=SCOD). This paper collects corresponding data from the following two aspects. First, the classification number in the patent authorized by the logistics enterprise belongs to the manufacturing industry. The specific methods are as follows: search the valid patents of logistics companies (the company name includes logistics, transportation, storage, freight, express, express, etc.) in the patent application (patent right) and then screen out the patents of logistics enterprises in the field of manufacturing based on the patent classification number. The patent classification number is based on the national IPC classification number. In the classification number of the patent applied by the logistics company, the codes B60-B67, E01, E06, F01-F28, H04, G01, G03, G09, G11, and G16 are extracted as screening conditions. Second, the patents authorized by manufacturing enterprises belong to the field of logistics. Search for valid patents for the keyword includes one of logistics, transportation, warehousing, freight, express, express, express, packaging, loading and moving in the patent subject. Then, the patents whose applicants are logistics enterprises were deleted, and the remaining patents were manually sorted out to select enterprises belonging to the manufacturing field, mainly selecting enterprise types with a high degree of connection with the logistics industry, such as machinery manufacturing, electrical appliances, equipment, electronics, communication and other enterprises. A total of 9,058 manufacturing enterprises were selected. The patents collected from both sides were summarized, and duplicates were removed. Finally, a total of 20,454 cross-field patents applied by the two industries from 2006 to 2020 that met both conditions were selected. To further test the accuracy of the collected data, we asked graduate students from the research team to quickly browse the summary content of 20,454 patents and conduct a secondary check. No unqualified patents were found. The data of the impact variables is sourced from the ESPDATA database (https://www.epsnet.com.cn).

### 3.2 Methodology

Geographic Information System (GIS) spatial analysis: As an emerging method, combining Geographic Information System can mine the data with Geographic Information to reveal the spatial significance that cannot be revealed by traditional statistical and econometric methods.

This paper uses ArcGIS software to analyse the exploratory spatial data of the collaborative innovation between the two industries.

Global spatial autocorrelation model: Global spatial autocorrelation reflects the overall degree of correlation in the spatial distribution of variables, usually using the global Moran's index to measure the degree of correlation between variable observations and their spatial lags.

Spatial weight matrix: In this paper, the geographical distance weight matrix is used for benchmark regression, which calculates the geographical distance between cities according to the longitude and latitude of the centroid of each city and is constructed by its reciprocal. The formula is:

$$W_{ij}^1 = R \times \arccos(\cos(x_i - x_j)\cos y_i \cos y_j + \sin y_i \sin y_j) \tag{1}$$

where $W_{ij}$ is the spatial weight matrix; $R$ is the equatorial radius of the earth, with a value of 6378 km; $x_i$, $x_j$, $y_i$, $y_j$ are the longitude and latitude of city $i$ and city $j$, respectively.

Spatial Dubin model: The spatial Dubin model is used to analyse the influencing factors of collaborative innovation between the two industries. The formula is:

$$\ln(Y_{it}) = \rho \sum\nolimits_{j=1}^{n} W_{ij}\ln(Y_{it}) + \beta\ln(X_{it}) + \sum\nolimits_{j=1}^{n} W_{ij}\ln(X_{it})\gamma + \mu_i + \lambda_t + \varepsilon_{it} \tag{2}$$

where $Y_{it}$ is the level of collaborative innovation between the two industries of city $i$ in year $t$; $X_{it}$ is the influencing factor of city $i$ in year $t$; $\rho$ is the spatial lag regression coefficient, which represents the impact of the local level of collaborative innovation between two industries on the level of collaborative innovation between two industries in neighboring cities; and $\gamma$ is the spatially lagged regression coefficient of independent variables. When $\gamma = 0$, the SDM model is the SLM model, and when $\gamma + \rho\beta = 0$, the SDM model is the SEM. $\mu_i$ represents the spatial fixed effect, and $\lambda_t$ represents the time fixed effect. $\varepsilon_{it}$ is the spatial autocorrelation error term, and $W_{ij}$ is the spatial weight matrix. $W_{ij}\ln(Y_{it})$ represents the interactive influence of $\ln(Y_{it})$ of city $i$ on $\ln(Y_{it})$ of neighboring cities; $W_{ij}\ln(X_{it})$ represents the interaction effect of $\ln(X_{it})$ with city $i$ on $\ln(X_{it})$ of neighboring city.

## 4. Analysis of evolution characteristics of collaborative innovation between logistics industry and manufacturing industry

### 4.1 Evolution stage

To analyse the evolution process and characteristics of collaborative innovation between the two industries, based on the number of collaborative innovation cities and the total number of patents from 2006 to 2020, this paper divides the evolution of collaborative innovation between the two industries into three stages: embryo, rapid development and steady development (see Fig 1).

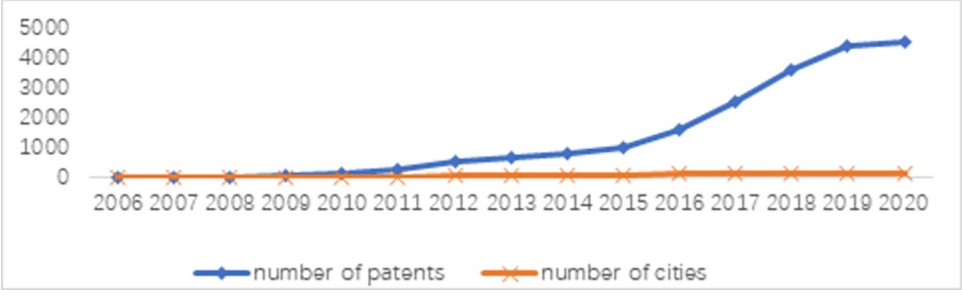

**Fig 1. Number of patents and cities of collaborative innovation.**

1. **Embryo stage (2006–2009)**

At this stage, the number of cities with collaborative innovation between the two industries is relatively low, and the total number of patents is low, which indicates that the participant is not active. In 2006, there were only 7 cities in which collaborative innovation between the two industries was created, and the maximum number of patents in a single city was only 4. Except for Guiyang, the other six cities were all first-tier cities, which indicated that the foundation for the joint development of the two industries was poor. In 2009, there were still more than 80% cities without collaborative innovation between the two industries.

1. **Rapid development stage (2010–2015)**

At this stage, both the number of cities and the total number of patents increased. In 2010, the Office of the National Modern Logistics Inter-Ministerial Joint Conference issued the "Opinions on Promoting the Joint Development of Manufacturing and Logistics Industry" and "Notice on Carrying out Demonstration Work for Joint Development of Manufacturing and Logistics Industry". The joint development of the two industries has become a key project supported by the government. The proportion of manufacturing logistics outsourcing has also increased significantly. In 2010, the proportion of outbound payment logistics costs of industrial, wholesale and retail enterprises was 52.5%, which exceeded half of the enterprise logistics costs for the first time. In 2020, the value increased to 69.7%.

1. **Steady development stage (2016–2020)**

From 2016 to 2020, the average annual growth rate of the total number of patents and the number of cities was 35.89% and 11.81%, respectively. In 2017, the number of cities where collaborative innovation occurred reached 144, more than half of the sample cities for the first time. In 2019, the growth rate of the number of cities and the total number of patents showed a slowing trend. In 2020, the number of cities with collaborative innovation was 177, and the maximum number of patents in a single city was 362 in Shanghai, which may be related to the outbreak of COVID-19 at the end of 2019.

## 4.2 Spatial distribution

In the embryo stage, the cities whose collaborative innovation quantity exceeds 50% of the maximum value in that year are concentrated in first-tier cities, with no obvious agglomeration characteristics in spatial distribution. Cities with the number of patents between 0 and 50% of the maximum value in the current year are few in the embryo stage, and their spatial distribution is scattered. In the stage of rapid development, Hangzhou, Suzhou and other places have been added to the cities with the number of collaborative innovation patents exceeding 50% of the maximum value of the year, and there is still no obvious agglomeration in spatial distribution. The number of cities with the number of collaborative innovation patents between 0 and 50% of the maximum value in that year increased the most, and the spatial distribution gradually spread from the eastern coast to the central region. In the stable development stage, the number and spatial distribution of cities that exceed 50% of the maximum value in the current year do not change significantly. Overall, the proportion of cities with patents between zero and 50% of the maximum value grew the fastest, which increased from 2.1% to 46.1% during the study period, while the proportion of city with patents of zero collaborative innovation patents decreased from 97.5% to 38.03%.

Table 1 shows the standard deviation ellipse of collaborative innovation in some years. In the embryo stage, the average value of the difference between the major and minor axes of the standard deviation ellipse is 1.95, and the average area of the ellipse is 162.42. In the rapid

**Table 1. Standard deviation ellipse for some years.**

| stage | year | shape-length | shape-area | center x | center y | XStdDist | YstdDist | Rotation |
|---|---|---|---|---|---|---|---|---|
| Embryo | 2006 | 45.91 | 166.43 | 113.26 | 28.63 | 6.78 | 7.82 | 33.14 |
| | 2009 | 45.98 | 158.54 | 116.04 | 30.42 | 5.81 | 8.68 | 41.77 |
| Rapid development | 2010 | 45.94 | 162.72 | 116.29 | 30.05 | 6.22 | 8.32 | 36.51 |
| | 2015 | 46.08 | 160.68 | 116.60 | 31.69 | 5.95 | 8.60 | 39.21 |
| Steady development | 2016 | 45.52 | 158.99 | 116.53 | 31.16 | 6.08 | 8.32 | 31.34 |
| | 2020 | 44.27 | 138.75 | 116.76 | 30.92 | 5.01 | 8.82 | 31.41 |

development stage, the average annual flattening rate of the standard deviation ellipse increases to 2.38, and the average annual area is 161.69. In the stable development stage, the flatness increases to 3.03, while the area decreases to 148.87. The trend of flattening indicates that the direction of collaborative innovation is becoming increasingly obvious. The reduction in the ellipse area indicates that the spatial agglomeration characteristics of collaborative innovation are more obvious. The spatial rotation angle of the standard deviation ellipse changes from 33.140˚ in 2006 to 31.406˚ in 2020, gradually shifting towards the east coast, and finally, the east–west direction is dominant. By 2020, the coverage of the standard deviation ellipse is basically consistent with that of the Yangtze River Delta urban agglomeration and the urban agglomeration in the middle reaches of the Yangtze River, indicating that these two urban agglomerations play important roles in the coordinated development of the two industries.

Table 2 shows the cold and hot spot analysis results of collaborative innovation between the two industries in the starting and ending years of the three development stages. As seen from the Table 2, the gravity of collaborative innovation between the two industries is relatively stable at the stage of development and stable development, basically stable in Anqing City, Anhui Province, with a trend of gradually shifting eastward and northward. At the embryonic stage, the city in the cold spot area is rare, and the spatial distribution is characterized by agglomeration. The number of cities in hot spots has increased significantly, and the spatial agglomeration area has shifted from the southwest to the east coast and the northern coastal areas. At the rapid development stage, there are not many cities in the cold spot area, and the spatial distribution is relatively scattered. The growth rate of the number of cities in hot areas has declined, and the spatial distribution is still concentrated in the eastern and southern coastal areas. At the stable development stage, the number of cities in the cold spot area increased, and the spatial distribution gradually spread to the south of the northwest and southwest regions. The

**Table 2. The cold and hot spot analysis results.**

| Stage | Year | Center of gravity | Cold spot area | Hot spot area |
|---|---|---|---|---|
| Embryo | 2006 | (E113˚26′, N28˚63′) Changsha city | 6 cities (agglomerations) Henan province | 19 cities (agglomerations) Southwest China |
| | 2009 | (E116˚04′, N30˚42′) Anqing city | 4 cities Border of Henan and Hubei province | 57 cities (agglomerations) Eastern and northern coasts |
| Rapid development | 2010 | (E116˚29′, N30˚05′) Anqing city | 0 city | 60 cities (agglomeration) Eastern and northern coasts |
| | 2015 | (E116˚60′, N31˚69′) Liu 'an city | 16 cities (scattered) | 73 cities (agglomeration) Eastern and northern coasts |
| Steady development | 2016 | (E116˚53′, N31˚16′) Anqing city | 24 cities (scattered) | 75 cities (agglomeration) Eastern and northern coasts |
| | 2020 | (E116˚76,N30˚92) Anqing city | 62 cities (agglomeration) The south of the northwest and southwest regions | 81cities (agglomeration) Eastern and northern coasts |

number of cities in hot spots is relatively stable, and the spatial distribution remains unchanged. On the whole, the collaborative innovation between the two industries presents the phenomenon of "less overall and partial concentration" in space, and the collaborative innovation in the eastern and western regions is unbalanced. The hot spots converge in East China, showing some spatial agglomeration characteristics. The cold-spot region has no obvious spatial agglomeration characteristics in the embryo and rapid development stages but shows certain spatial agglomeration characteristics in the stable development stages.

In the evolution of the spatial structure of collaborative innovation between the two industries, the innovation level, innovation policy support and multidimensional proximity of the region play very important roles. In terms of innovation level, Shanghai, a city in the Yangtze River Delta, was in the leading position in the number of collaborative innovation between the two industries during the study period. The number of collaborative innovation patents in Hangzhou, Suzhou, Nanjing, Wuxi and other cities also increased significantly in the rapid and stable development stages. In 2020, the Yangtze River Delta urban agglomeration contributed approximately 25% of the country's total GDP, with an area of 2.3%. It has nearly 25% of the country's "double first-class" universities, and the number of effective inventions and R&D expenditures account for approximately 33% of China. According to the data from the "Innovative Enterprise Database", the number of innovative manufacturing enterprises in Zhejiang and Jiangsu was higher than that in Guangdong, Beijing and Tianjin in 2018. In terms of the internal expenditure of R&D funds and the amount of R&D personnel, Jiangsu, Shanghai and Zhejiang ranked among the top six all year round. In 2019, the "China Regional Innovation and Entrepreneurship Index" released by the enterprise big data research center of Peking University, Zhejiang, Jiangsu and Shanghai in the Yangtze River Delta ranked second, third and fifth, respectively, in terms of total scores. The overall innovation level of the Yangtze River Delta region is relatively high. In terms of innovation policy, the Yangtze River Delta urban agglomeration actively creates a good environment for collaborative innovation. In 2020, the Ministry of Science and Technology issued the "Construction and Development Plan for the Yangtze River Delta Science and Technology Innovation Community" to enhance the collaborative innovation capacity of the Yangtze River Delta region. In conclusion, the Yangtze River Delta urban agglomeration has the advantages of geographical proximity, institutional proximity and economic proximity in terms of collaborative innovation between the two industries, while collaborative innovation is more likely to occur in the urban agglomeration.

## 5. Analysis of the influencing factors of collaborative innovation development between the two industries

### 5.1 Variable selection

Referring to the existing related studies [45, 46], this paper selects economic development, science and technology, government behavior, employment population, informatization level, and logistics infrastructure as the influencing factors of collaborative innovation between the two industries. The representative indicators and descriptive statistics of each factor are shown in Table 3.

Economic development: Regions with high levels of economic development have a greater demand for new technologies, prompting various enterprises to strengthen their mastery, utilization, and absorption of heterogeneous resources. Therefore, they are more inclined to seek partners to support innovation activities [23], which helps promote the generation of collaborative innovation between the two industries. On the other hand, regions with high levels of economic development are rich in innovation resources, and innovation entities are less constrained by heterogeneous resources, which may reduce their willingness to cooperate.

Table 3. Variable definition and descriptive statistics.

| Factor | Definition | Variable | Mean | Max | Min | SD |
|---|---|---|---|---|---|---|
| Collaborative innovation between the two industries | Patents authorized by the two industries in each other's fields (PCS) | lnY | 0.53 | 5.97 | 0 | 1.06 |
| Economic development | Per capita gross domestic product(Yuan) | lnP | 10.44 | 13.06 | 4.6 | 0.72 |
| Science and technology | Ratio of scientific expenses to GDP (%) | SCI | 0.02 | 0.63 | 0.0002 | 0.02 |
| Government behavior | Ratio of public finance expenditure to GDP (%) | GOV | 0.18 | 1.49 | 0.01 | 0.11 |
| Employment population | The number of employees of public institutions in urban (in 10,000 people) | lnE | 3.54 | 7.04 | 1.44 | 0.84 |
| Informatization level | The number of internet broadband access users(in 10,000 people) | lnI | 4.29 | 16.50 | -1.25 | 2.86 |
| Logistics infrastructure | Road mileage(kilometer) | lnR | 9.18 | 12.11 | 4.44 | 0.73 |

Therefore, the level of economic development may also limit the generation of collaborative innovation between the two industries.

Science and technology: In regions with high technological levels, the more developed their technology markets are, the higher the demand for technology will be, and therefore, cooperation among various innovation entities will be promoted. However, in regions with high technological levels, where there are abundant categories of technological products on the market, it is also possible to reduce the difficulty of enterprises in obtaining heterogeneous technologies, thereby reducing the willingness of enterprises to innovate and cooperate between enterprises and across industries.

Government behavior: The government can encourage collaborative innovation through financial support, research and development subsidies, and other forms [47]. However, the involvement of government funds and their leverage effect can also alleviate the scarcity of resources to a certain extent, reducing the willingness of innovation entities to cooperate, and may also be difficult to achieve an ideal promotion effect on industry-university-research collaborative innovation [48].

Employment population: China's manufacturing industry is still dominated by labor intensive industries such as processing and assembly, and is highly dependent on the employment population [49]. China's logistics industry has also been a labor intensive industry, with a strong dependence on labor resources [50]. The employment population has a direct impact on the development of both industries, which in turn affects the collaborative innovation of the two industries.

Informatization level: The improvement of information technology such as the Internet can promote communication between innovation entities, facilitate the generation of collaborative innovation willingness among entities, and ultimately promote their development [51]. However, it may also further lead to excessive concentration of innovation factors, which is not conducive to the development of collaborative innovation [52].

Logistics infrastructure: Infrastructure construction is one of the evaluation indicators of logistics performance [53]. The improvement of transportation infrastructure can break the constraints of spatial and geographical location on innovative resource elements, promote the flow of elements between regions, and positively promote collaborative innovation between the two industries [54].

## 5.2 Spatial autocorrelation test

As seen from the previous analysis, collaborative innovation between the two industries shows a certain agglomeration feature in space. This section uses Moran's index to make a preliminary judgment on the spatial autocorrelation of collaborative innovation between the two

**Table 4. Moran's index test results.**

| Year | Moran's I | Z value | Year | Moran's I | Z value |
|------|-----------|---------|------|-----------|---------|
| 2006 | -0.002 | 0.351 | 2014 | 0.031 | 5.833 |
| 2007 | -0.001 | 0.391 | 2015 | 0.036 | 6.641 |
| 2008 | -0.001 | 0.908 | 2016 | 0.031 | 5.857 |
| 2009 | 0.008 | 2.000 | 2017 | 0.024 | 4.639 |
| 2010 | 0.008 | 2.182 | 2018 | 0.034 | 6.436 |
| 2011 | 0.006 | 1.653 | 2019 | 0.031 | 5.813 |
| 2012 | 0.003 | 5.730 | 2020 | 0.103 | 11.112 |
| 2013 | 0.021 | 4.177 | | | |

industries, and the results are shown in Table 4. As can be seen from Table 4, except for 2006–2008, the Moran's index of the level of collaborative innovation between the two industries in the other years is greater than 0, showing a fluctuating upward trend as a whole, and has passed the significance test, indicating that there is an obvious spatial autocorrelation between the two industries' collaborative innovation.

## 5.3 Regression result analysis

Table 5 shows the test results of the spatial econometric model. Both the spatial lag and spatial error pass the test. At the same time, the LR test also rejects the original hypothesis and passes the Hausman test. Therefore, the fixed-effect spatial Dubin model is selected for the next analysis [55].

Model 1 in Table 6 gives the regression results of the factors affecting the collaborative innovation between the two industries under the weight of geographical distance. The direct effect represents the impact of local influencing factors on local collaborative innovation between the two industries. The indirect effect is the impact of local factors on the collaborative innovation between the two industries in the surrounding areas.

As seen from Table 6, economic development has a positive effect on the collaborative innovation between the two industries in the local area. Economic development is an important guarantee of innovation investment. By comparing the proportion of R&D investment, it can be seen that the proportion of R&D investment in the total national economic output is relatively high in regions with a higher level of economic development. Guangdong province, Jiangsu province and Shandong province ranked the top three in GDP in 2019, and Guangdong province, Jiangsu province and Beijing city ranked the top three in internal R&D expenditure in the same period. The level of economic development has a significant negative impact on the collaborative innovation between the two industries in neighboring cities. The possible reason is that the improvement of the local economic level makes the local collaborative innovation development between the two industries form a certain advantage, resulting in a preference for innovation. In addition, it also has a negative externality on the collaborative innovation between the two industries in the neighboring cities.

**Table 5. The test result of the spatial econometric model.**

| Test | Z value | P value |
|------|---------|---------|
| LM(lag) test | 7.033 | 0.00 |
| Robust LM(lag) test | 52.78 | 0.00 |
| LM(error) test | 6.25 | 0.00 |
| Robust LM(error) test | 57.99 | 0.00 |

**Table 6. Results of spatial Dubin regression results.**

| Effect | Variable | Model 1 | Model 2 | Model 3 |
|---|---|---|---|---|
| Direct effect | lnP | 0.400*** (5.253) | 0.748*** (12.713) | 0.557*** (7.212) |
| | SCI | 0.035*** (5.963) | 0.048*** (7.739) | 0.018* (1.984) |
| | GOV | 0.076** (1.792) | 0.010 (0.276) | 0.223** (2.332) |
| | lnE | 0.345*** (6.429) | 0.338*** (6.261) | 0.427* (1.770) |
| | lnI | -0.304*** (-9.982) | 0.062*** (2.838) | -0.174** (-2.311) |
| | lnR | -0.103** (-2.394) | -0.114*** (-2.640) | -0.202(-1.622) |
| | SEC | | | 0.005***(3.340) |
| Indirect effect | lnP | -2.403* (-1.70) | -134.339* (-1.819) | 0.005***(3.340) |
| | SCI | 12.207* (1.682) | 11.331* (1.791) | -3.149*(-1.607) |
| | GOV | -26.095 (-1.467) | -19.185 (-0.529) | 10.542*(1.891) |
| | lnE | 26.095 (1.197) | -49.092 (-0.618) | -18.025*(-1.722) |
| | lnI | 16.799 (1.570) | 13.206 (0.757) | 24.233(0.742) |
| | lnR | 7.035 (0.524) | -83.477 (-0.948) | 14.466(0.817) |
| | SEC | | | 0.022(0.05) |

Note: ***, ** and * represent significant at 1%, 5% and 10% statistical levels, respectively.

The promotion of science and technology on the collaborative innovation between the two industries in the local and neighboring cities is significant. High and new technologies are increasingly widely used in enterprises, which improves the level of intelligence and information of enterprises, reduces the barriers of cooperation and communication between enterprises, and promotes the development of collaborative innovation among enterprises. On the other hand, the increase in the proportion of science and technology investment will directly promote logistics enterprises and manufacturing enterprises to increase the input of innovation factors and generate more innovative behaviors, which is bound to increase the occurrence probability of collaborative innovation between the two industries.

Government behavior has a significant positive impact on local collaborative innovation between the two industries. Government behavior can affect the location selection and spatial layout of the logistics and manufacturing industries through various land use policies, logistics (industrial) park development plans, and other special plans. At the same time, financial subsidies can significantly increase the number of technological innovations of enterprises [56], which further affects the level of collaborative innovation between the two industries. The local government supports and encourages local enterprises to actively innovate by means of financial subsidies and tax incentives, which reduces the risks and costs of enterprise innovation and provides a good foundation for collaborative innovation between the two industries.

The employment population has a significant positive impact on the local collaborative innovation between the two industries. In China, manufacturing and logistics are labor-intensive industries. In 2020, the employed population in the logistics industry accounted for 4.77% of the total employed population, and that of the manufacturing industry was 22.33%, ranking seventh and first among all industrial employed populations, respectively. Therefore, labor resources play a positive role in promoting collaborative innovation between the two industries.

The level of informatization has a significant negative impact on local collaborative innovation between the two industries. The possible reason is that although the development of informatization has facilitated the exchange and communication between local innovation entities, it has not established a real effective connection and failed to effectively enhance the strength of the connection between the two industries. The development of information technology

also makes it easier to obtain information, which may increase the gap in information acquisition and the asymmetry of information. For logistics companies or manufacturing companies that are disadvantaged in information resources, the efficiency of their allocation of innovation elements will be reduced due to information asymmetry, resulting in damage to the enthusiasm for external innovation contacts [57].

Logistics infrastructure has a significant negative impact on the local collaborative innovation between two industries. The possible reason is that the improvement of the logistics infrastructure has reduced the time and distance cost of external communication for enterprises, and it is easier for logistics enterprises and manufacturing enterprises to cooperate across regions and borders, which may reduce the intraregional cooperation, thereby hindering the collaborative innovation between the two industries in the local area.

## 5.4 Robustness test

This paper uses the following two methods to test the robustness of the empirical results.

(1) Change the space weight matrix. In two regions with different levels of economic development, the development level of logistics and manufacturing industries is significantly different, which will lead to different levels of collaborative innovation between the two industries. In this paper, the economic distance weight matrix is used to replace the geographical distance weight matrix for robustness test. The estimated results of the model are shown in Model 2 in Table 6. The calculation formula of economic distance weight matrix is as follows:

$$W_{ij}^2 = 1/|\overline{GDP_i} - \overline{GDP_j}| \tag{3}$$

where $\overline{GDP_i}$ and $\overline{GDP_j}$ are the mean per capita GDP of cities $i$ and $j$, respectively.

(2) Add control variables. The regional industrial structure has a direct impact on the demand of logistics industry, especially the demand of the secondary industry and the tertiary industry for logistics industry is greater than that of the primary industry. The manufacturing industry itself belongs to the secondary industry. Therefore, the proportion of the secondary industry in the region will directly affect the development of the logistics industry and manufacturing industry, and then affect the collaborative innovation between the two industries. This paper chooses the proportion for robustness test of added value of the secondary industry to GDP to represent the industrial structure. The estimated results of the model are shown in Model 3 in Table 6.

The estimation results of Model 2 in Table 4 show that under the weight of economic distance, the impact direction of information technology level on local collaborative innovation between two industries, and the impact direction of employment population on collaborative innovation between two industries in surrounding areas are inconsistent with model 1. Under the weight of economic distance, the level of informatization has significantly promoted the development of collaborative innovation between the two industries in local. The possible reason is that the development of informatization has weakened the hindering effect of spatial distance on collaborative innovation. With the improvement of the information technology level, the spatial distance of collaborative innovation between the two industries is lengthened. Cooperation between the two industries is more about mutual benefit of economic development than geographical proximity. The regression results of other influencing variables are basically consistent with those of Model 1, except that the coefficient size and significance level are somewhat different. The estimated result of model 3 is basically consistent with that of model 1, but only the difference of coefficient size and significance level. On the whole, the empirical results of the analysis of the factors affecting the development of collaborative innovation between the two industries are relatively stable.

## 6. Conclusions and suggestions

### 6.1. Conclusions

Collaborative development between the two industries has been a hot topic in recent years, but there are few research results on collaborative innovation between the two industries. In this paper, the cross-field authorized patents between the two industries are taken as the measurement index of collaborative innovation, and the evolution characteristics and influence factors are systematically and comprehensively described. The following research conclusions are obtained:

1.  From 2006 to 2020, the number of cities and the total number of patents for collaborative innovation between the two industries in China increased significantly. However, the number of cities without collaborative innovation accounted for a relatively large proportion, while the number of cities with collaborative innovation exceeding 50% of the maximum number of patents in that year was very small. Overall, the level of collaborative innovation in China's two industries is not high. In terms of spatial distribution, cities with strong collaborative innovation capabilities are clustered into a few first-tier cities along the eastern coast, and the blank areas of collaborative innovation are clustered in the western. In terms of time distribution, the collaborative innovation between the two industries mainly experienced the embryo stage (2006–2009), rapid development stage (2010–2015), and stable development stage (2016–2020).

2.  The spatial distribution pattern of collaborative innovation between the two industries is dominated by the north-south direction. The urban agglomeration in the Yangtze River Delta and the middle reaches of the Yangtze River play important roles in collaborative innovation between the two industries. The hot spots and subhot spots of collaborative innovation gradually shifted from the Pearl River Delta region to the eastern coastal region, and the cold spots gradually spread from the central region to the western region. The hot and cold areas have a certain clustering trend. There is no obvious tendency for the cold spots to become hot or subhot spots.

3.  Among the influencing factors, economic development, science and technology, government behavior, and employment population are positive influencing factors for the collaborative innovation development between the two industries on the local, while information technology level and logistics infrastructure are negative influencing factors. The local economic development will have a negative spatial effect on the level of collaborative innovation between the two industries in the surrounding area, while the local scientific and technology will have a positive spatial spillover effect on the level of collaborative innovation between the two industries in the surrounding area.

### 6.2 Suggestions

Based on the above research conclusions, this paper puts forward the following suggestions for the sake of the improvement of collaborative innovation ability between the two industries.

1.  It is necessary to give full play to the radiation power of hotspot areas, strengthen cooperation between regions, and build a collaborative innovation network with open sharing and complementary resources. Advanced information technology should be used to strengthen the cross-field and cross-regional communication between the two industries and enhance the innovation spillover effect of the Yangtze River Delta and the middle reaches of the Yangtze River urban agglomeration on other regions to weaken the polarization

phenomenon of collaborative innovation between the two industries. Taking urban agglomeration as the carrier, we should focus on cultivating the construction of collaborative innovation between two industries within and among urban agglomerations in the eastern region. We will improve the collaborative innovation environment in the central and western regions by vigorously developing the economy and increasing investment in scientific research.

2. Judging from the results of influencing factors, it is necessary to improve the regional economic development level and build a market-led, enterprise-based, government-assisted collaborative innovation market system with a strong economic level to promote logistics market demand growth to further promote collaborative innovation between the two industries. Increase investment in R&D and government financial expenditure to create a good environment for innovation. The government can encourage logistics companies and manufacturing companies to build a common technology research and development platform through subsidies or tax relief. Technological cooperation between the two industries should be encouraged to increase the flow and sharing of innovative elements. Attention should be given to the introduction and cultivation of talent. The local government should establish a reasonable mechanism for the cultivation, introduction and training of talent and create a good employment environment. Encourage the logistics enterprises and the manufacturing enterprises to take advantage of the "new infrastructure" to meet the individual demand of the market with infrastructure innovation and facilitate the development of collaborative innovation between the two industries.

This paper takes the collaborative innovation between the two industries as the research object, explores the characteristics and influencing factors of the evolution of collaborative innovation between the two industries, and provides theoretical reference for promoting the level of collaborative innovation between the two industries. However, because the content of data collected is not rich enough, there is no way to further analyze the heterogeneity of manufacturing industry types. In the future, we can further investigate the heterogeneity, and whether there are intermediary effects or threshold effects of individual impact variables on the development of collaborative innovation between the two industries, such as the level of economic development, industrial agglomeration, etc.

## Supporting information

**S1 File.**
(XLSX)

## Author Contributions

**Conceptualization:** Haohua Liu.

**Data curation:** Haohua Liu.

**Formal analysis:** Xiuling Chen.

**Funding acquisition:** Haohua Liu.

**Investigation:** Haohua Liu.

**Methodology:** Xiuling Chen.

**Project administration:** Haohua Liu.

**Resources:** Haohua Liu.

**Software:** Xiuling Chen.

**Validation:** Xiuling Chen.

**Visualization:** Xiuling Chen.

**Writing – original draft:** Xiuling Chen.

**Writing – review & editing:** Xiuling Chen, Haohua Liu.

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
