## [Decision Letter · Decision Letter 0]

20 Feb 2023

PONE-D-23-00850Collaborative Innovation Evolution of Logistics and Manufacturing industry in ChinaPLOS ONE

Dear Dr. Xiuling,

Thank you for submitting your manuscript to PLOS ONE. After careful consideration, we feel that it has merit but does not fully meet PLOS ONE’s publication criteria as it currently stands. Therefore, we invite you to submit a revised version of the manuscript that addresses the points raised during the review process.

We look forward to receiving your revised manuscript.

Kind regards,

László Vasa, PhD

Academic Editor

PLOS ONE

Journal Requirements:

4. We note that Figures 2 and 3 in your submission contain [map/satellite] images which may be copyrighted. All PLOS content is published under the Creative Commons Attribution License (CC BY 4.0), which means that the manuscript, images, and Supporting Information files will be freely available online, and any third party is permitted to access, download, copy, distribute, and use these materials in any way, even commercially, with proper attribution. For these reasons, we cannot publish previously copyrighted maps or satellite images created using proprietary data, such as Google software (Google Maps, Street View, and Earth). For more information, see our copyright guidelines: http://journals.plos.org/plosone/s/licenses-and-copyright.

a) You may seek permission from the original copyright holder of Figure(s) [#] to publish the content specifically under the CC BY 4.0 license.  

Natural Earth (public domain): http://www.naturalearthdata.com/.

Reviewers' comments:

Reviewer's Responses to Questions

**Comments to the Author**

1. Is the manuscript technically sound, and do the data support the conclusions?

Reviewer #1: Yes

Reviewer #2: Yes

2. Has the statistical analysis been performed appropriately and rigorously? 

Reviewer #1: Yes

Reviewer #2: Yes

3. Have the authors made all data underlying the findings in their manuscript fully available?

Reviewer #1: Yes

Reviewer #2: Yes

4. Is the manuscript presented in an intelligible fashion and written in standard English?

Reviewer #1: Yes

Reviewer #2: Yes

5. Review Comments to the Author

Reviewer #1: Comments and Suggestions for Authors

First of all, I appreciate the opportunity to review your paper, which title is:

Collaborative Innovation Evolution of Logistics and Manufacturing industry in China

The topic is very interesting and important. The subject is significant and relevant and the study contributes to the development of scientific knowledge. The problem presented in relation to the results is very useful, however, the article could be improved on several points:

Title: The title is clear, expressive and acceptable to me.

Abstract is well structured and more or less clear, but there are some missing parts. Rather, focus on the importance of the topic and the relevant (or main) question. The research objectives are not defined, the scientific gaps and the goal are missing. The wording of the second result is not clear and a final conclusion is also missing.

Introduction is well-structured, clear and well-written at the present form. The first impression of reading the introduction is that it is brief but informative and focusing on the narrowly defined topic. However, the introduction should focus much more clearly on the importance of the topic, research goals and scientific gaps, correlating context and purpose. It should be more expressive and comprehensive. In addition in the introduction some new literature sources should be cited especially in the following areas.

a) Main scientific issues and latest debates in the field, scientific uncertainties and gaps between the theories. I suggest incorporating prestigious international journals with foreign authors.

The literature review. The literature review is moderate and adequately supports the topic.

Data sources and methodology:

The methodology is clear, but about sample selection you should add some more important information about database, how and where it can be found even if it is not public for everyone. Some reference is needed. Secondly, please explain what “other statistics” mean.

Please correct or explain the followings:

a) Second equation is not well edited.

b) Please cite the spatial Dubin model correctly and Moran's index too.

c) Please use Arabic letters for the database city names.

Research findings (Conclusions and suggestions)

The conclusions are well structured, but the limitations of the model and future research directions are lacking. Please provide further suggestions for generalizing the model and results. How can stochastic factors affect the presented result and which factors can affect the uncertainty behaviour?

The paper has good potential, but the authors need to make a minor revision at this stage. I’m very glad to have the opportunity to read your work.

Recommendation: Minor Revisions Required

Reviewer #2: Review report for the paper “Collaborative Innovation Evolution of Logistics and Manufacturing industry in China”

The applicability of the method. Why do we need application spatial modelling in this study? I did not see the author discussing the reason. Therefore, it is impossible to prove the superiority of this model combination in this article. Need detailed further explanation.

Insufficient expression on innovative explanations. Does the practical significance of this innovation exist? There is a lack of comparison with previous studies of the same kind. For this point, the innovativeness of the author's statement needs further explanation.

Indicator issues. Is it appropriate for the author to directly use research results in similar literatures into the research questions of this article? Is there a better reference standard in similar studies? In the subdivision question of this article, do you need to further improve the research results of other scholars in the index design? Please give a reliable argument for the indicator design.

Literature review. Add more recent papers published in last three years. Remove papers published before 2018. Based on the LR you should define the scientific gap. I suggest authors to read and discuss following papers: Karamaşa, Çağlar, Demir, E., Memiş, S., & Korucuk, S. (2021). Weighting the factors affectıng logıstıcs outsourcıng. Decision Making: Applications in Management and Engineering, 4(1), 19-32; Stojanović, I., & Puška, A. (2021). Logistics Performances of Gulf Cooperation Council’s Countries in Global Supply Chains. Decision Making: Applications in Management and Engineering, 4(1), 174-193..

Model selection problem. The research question of this article, does the proposed methodology get better results than the old/existing methods? There is no comparative proof, no analysis of the superiority of the method. Lack of comparison of results under different models. Not that the new method is equally applicable to all problems.

There is no result robustness. The author needs to give more detailed data references or results.

In the part of research status, the outline of the whole research is not clear enough, and more content of multi criteria decision model (method) needs to be added.

The results of the application part of the model need to be rearranged, the readability is too poor, and the graphical results provided can’t make people see the differences under different scene settings.

6. PLOS authors have the option to publish the peer review history of their article (what does this mean?). If published, this will include your full peer review and any attached files.

Reviewer #1: **Yes: **Dr. Gyenge Balázs

Reviewer #2: No

---

## [Author Response · Author response to Decision Letter 0]

6 Apr 2023

Response to Reviewers

Response to Reviewer#1

Thank you for your insightful comments and careful guidance. We have now carefully analyzed and made modifications based on your feedback. We have revised the abstract and introduction of the paper, adding the importance, research gaps, and significance of the paper. Removed older literature and added literature from the past three years. The introduction section reiterates the innovative points of this paper based on the focus of this study. We have increased the limitations of our research and future research directions. After receiving your valuable feedback, our article level has greatly improved.

Response to Reviewer#2

Thank you for your insightful comments and careful guidance. Now we have carefully analyzed and revised exactly according to your comments. We have added explanation of the applicability of the method. The introduction and literature review sections have been significantly revised to enhance the persuasiveness of this article's innovation. At the same time, the cited literature has been updated, deleting some literature before 2018 and adding some literature from the past three years. Added explanation of influencing factor indicators. Added explanation for using spatial measurement methods. Robustness test is added. The research status has been reorganized. The image display of the results has been modified into a table. Thank you for your suggestion to improve the quality of the paper.

Comment 1（Reviewer #1）: Title: The title is clear, expressive and acceptable to me. Abstract is well structured and more or less clear, but there are some missing parts. Rather, focus on the importance of the topic and the relevant (or main) question. The research objectives are not defined, the scientific gaps and the goal are missing. The wording of the second result is not clear and a final conclusion is also missing.

Response 1: The abstract of the paper has been revised. The importance of the topic, the research objectives, the scientific gaps and the goal are all included in it.

The research importance is as follows: In the increasingly fierce market competition, open collaborative innovation is more conducive to improving the level of linkage between the logistics industry and the manufacturing industry, and promoting industrial development.

Objectives of the study: This article aims to explore the current situation and influencing factors of collaborative innovation between the two industries, with a view to proposing countermeasures and suggestions for improving the level of collaborative innovation between the two industries, and also providing new ideas for research on cross industry collaborative innovation. 

Scientific gaps and the goal: This article aims to explore the current situation and influencing factors of collaborative innovation between the two industries, with a view to proposing countermeasures and suggestions for improving the level of collaborative innovation between the two industries, and also providing new ideas for research on cross industry collaborative innovation.

Comment 2（Reviewer #1）：Introduction is well-structured, clear and well-written at the present form. The first impression of reading the introduction is that it is brief but informative and focusing on the narrowly defined topic. However, the introduction should focus much more clearly on the importance of the topic, research goals and scientific gaps, correlating context and purpose. It should be more expressive and comprehensive. In addition in the introduction some new literature sources should be cited especially in the following areas.

a) Main scientific issues and latest debates in the field, scientific uncertainties and gaps between the theories. I suggest incorporating prestigious international journals with foreign authors.

The literature review. The literature review is moderate and adequately supports the topic.

Data sources and methodology: The methodology is clear, but about sample selection you should add some more important information about database, how and where it can be found even if it is not public for everyone. Some reference is needed. Secondly, please explain what “other statistics” mean.

Please correct or explain the followings:

a) Second equation is not well edited.

b) Please cite the spatial Dubin model correctly and Moran's index too.

c) Please use Arabic letters for the database city names. 

Response 2: The introduction has been significantly revised, gradually transitioning from existing scientific issues and the latest discussions to the research topic of this article, highlighting the importance and innovation of this study. In the literature section, some ancient literature has been deleted and the latest three years' literature has been added.Added the source website link of the database, and other statistical data refers to the data sources of influencing factor variables, which have been modified and explained in the paper.

The second equation was modified.

The Dubin model and Moran's index were modified.

Use Arabic letters for the database city names.

Comment 3（Reviewer #1）：Research findings (Conclusions and suggestions)

The conclusions are well structured, but the limitations of the model and future research directions are lacking. Please provide further suggestions for generalizing the model and results. How can stochastic factors affect the presented result and which factors can affect the uncertainty behavior?

Response 3: The limitations of the research and future research directions have been added, as follows: This paper takes the collaborative innovation between the two industries as the research object, explores the characteristics and influencing factors of the evolution of collaborative innovation between the two industries, and provides theoretical reference for promoting the level of collaborative innovation between the two industries. However, because the content of data collected is not rich enough, there is no way to further analyze the heterogeneity of manufacturing industry types. In the future, we can further investigate the heterogeneity, and whether there are intermediary effects or threshold effects of individual impact variables on the development of collaborative innovation between the two industries, such as the level of economic development, industrial agglomeration, etc.

Comment 4（Reviewer #2）: The applicability of the method. Why do we need application spatial modelling in this study? I did not see the author discussing the reason. Therefore, it is impossible to prove the superiority of this model combination in this article. Need detailed further explanation.

Response 4: Added explanation on the applicability of the method. The details are as follows (see lines 195-212 of the paper): In the study of influencing factors of collaborative innovation, traditional quantitative analysis methods are commonly used, including the least square method (Hong J, et al., 2019) [8], the Tobit model (He L, et al., 2021) [38], and the negative binomial regression model (Chen H J, Xie F J, 2021; Hu S Y, Wu H C, 2021) [39,40]. Other methods such as structural equation modeling (Han K, 2021; Chen W, et al., 2020) [41,42], and social network analysis (Su Y, Cao Z, 2022) [43] are also involved. There are few literatures using spatial econometric models. Spatial econometrics is a popular method in regional economics, economic geography, and other fields in recent years, which can effectively evaluate the spatial effects of economic activities. Regarding the impact of spatial factors on collaborative innovation, many articles have included geographical distance as an influencing factor in traditional regression models (Xia L J, et al., 2017; Miguelez E., 2019) [44,45], but this treatment can only assess the impact of spatial factors on collaborative innovation, and cannot assess the spatial effects of other influencing factors. This article uses the spatial econometric method to analyze the influencing factors of collaborative innovation, which can not only evaluate the impact of local influencing factors on local collaborative innovation between the two industries, but also evaluate the impact of local influencing factors on collaborative innovation between the two industries in surrounding areas.

Comment 5（Reviewer #2）: Insufficient expression on innovative explanations. Does the practical significance of this innovation exist? There is a lack of comparison with previous studies of the same kind. For this point, the innovativeness of the author's statement needs further explanation.

Response 5: Significant modifications have been made in the literature review section. The practical significance of innovation is reflected in the following:

In China, the proportion of logistics outsourcing in industrial, wholesale and retail enterprises has reached 69.7% in 2020, which indicates that a considerable part of manufacturing enterprises have not outsourced their logistics business and belong to self-operated logistics. The activities of self-operated logistics of these manufacturing enterprises belong to the logistics industry functionally, but they are dominated by manufacturing enterprises. The innovation activities and achievements of these manufacturing enterprises in self-operated logistics are still registered in the name of the manufacturing enterprises when the patent application is made, but these innovation achievements belong to the logistics industry in terms of content. Considering the remaining nearly 30% of self-operated logistics, the collaborative innovation studied in this article includes not only the innovative activities of logistics enterprises and manufacturing enterprises as two independent individuals that cooperate with each other, but also the innovative activities of manufacturing enterprises in the logistics field alone, and the innovative activities of logistics enterprises in the manufacturing field alone. This may more comprehensively depict the situation of collaborative innovation between logistics and manufacturing industries in China.

Regarding the impact of spatial factors on collaborative innovation, many articles have included geographical distance as an influencing factor in traditional regression models (Xia L J, et al., 2017; Miguelez E, 2019)[44,45], but this treatment can only assess the impact of spatial factors on collaborative innovation, and cannot assess the spatial effects of other influencing factors. This article uses the spatial econometric method to analyze the influencing factors of collaborative innovation, which can not only evaluate the impact of local influencing factors on local collaborative innovation between the two industries, but also evaluate the impact of local influencing factors on collaborative innovation between the two industries in surrounding areas.

Comment 6（Reviewer #2）：Indicator issues. Is it appropriate for the author to directly use research results in similar literatures into the research questions of this article? Is there a better reference standard in similar studies? In the subdivision question of this article, do you need to further improve the research results of other scholars in the index design? Please give a reliable argument for the indicator design.

Response 6: The illustrative argument of the indicator is added. The details are as follows：

Economic development：Regions with high levels of economic development have a greater demand for new technologies, prompting various enterprises to strengthen their mastery, utilization, and absorption of heterogeneous resources. Therefore, they are more inclined to seek partners to support innovation activities (Ketokivi M, 2016) [24], which helps promote the generation of collaborative innovation between the two industries. On the other hand, regions with high levels of economic development are rich in innovation resources, and innovation entities are less constrained by heterogeneous resources, which may reduce their willingness to cooperate. Therefore, the level of economic development may also limit the generation of collaborative innovation between the two industries.

Science and technology：In regions with high technological levels, the more developed their technology markets are, the higher the demand for technology will be, and therefore, cooperation among various innovation entities will be promoted. However, in regions with high technological levels, where there are abundant categories of technological products on the market, it is also possible to reduce the difficulty of enterprises in obtaining heterogeneous technologies, thereby reducing the willingness of enterprises to innovate and cooperate between enterprises and across industries.

Government behavior: The government can encourage collaborative innovation through financial support, research and development subsidies, and other forms (Yoon J., Park H W., 2017) [48]. However, the involvement of government funds and their leverage effect can also alleviate the scarcity of resources to a certain extent, reducing the willingness of innovation entities to cooperate, and may also be difficult to achieve an ideal promotion effect on industry-university-research collaborative innovation (Marcelo C K., et al., 2017) [49].

Employment population: China's manufacturing industry is still dominated by labor intensive industries such as processing and assembly, and is highly dependent on the employment population (Luo Jun, 2019) [50]. China's logistics industry has also been a labor intensive industry, with a strong dependence on labor resources (Li Panke, 2020) [51]. The employment population has a direct impact on the development of both industries, which in turn affects the collaborative innovation of the two industries.

Informatization level: The improvement of information technology such as the Internet can promote communication between innovation entities, facilitate the generation of collaborative innovation willingness among entities, and ultimately promote their development (Zhang Jie, Fu Kui, 2021) [52]. However, it may also further lead to excessive concentration of innovation factors, which is not conducive to the development of collaborative innovation (Fang Gang, Tan Jiaxin, 2020)[53].

Logistics infrastructure: Infrastructure construction is one of the evaluation indicators of logistics performance (Ilija Stojanovi ć, Adis Pu š ka,2021[54].The improvement of transportation infrastructure can break the constraints of spatial and geographical location on innovative resource elements, promote the flow of elements between regions, and positively promote collaborative innovation between the two industries (Lei Shuzhen et al., 2021) [55].

Comment 7（Reviewer #2）：Literature review. Add more recent papers published in last three years. Remove papers published before 2018. Based on the LR you should define the scientific gap. I suggest authors to read and discuss following papers: Karamaşa, Çağlar, Demir, E., Memiş, S., & Korucuk, S. (2021). Weighting the factors affectıng logıstıcs outsourcıng. Decision Making: Applications in Management and Engineering, 4(1), 19-32; Stojanović, I., & Puška, A. (2021). Logistics Performances of Gulf Cooperation Council’s Countries in Global Supply Chains. Decision Making: Applications in Management and Engineering, 4(1), 174-193.

Response 7: The literature review section has undergone significant revisions. Added literature papers published in last three years. Remove papers published before 2018. The two literature suggested by experts have been read and cited.

Comment 8（Reviewer #2）：Model selection problem. The research question of this article, does the proposed methodology get better results than the old/existing methods? There is no comparative proof, no analysis of the superiority of the method. Lack of comparison of results under different models. Not that the new method is equally applicable to all problems. There is no result robustness. The author needs to give more detailed data references or results.

Response 8: Advantages of spatial measurement methods: This article uses the spatial econometric method to analyze the influencing factors of collaborative innovation, which can not only evaluate the impact of local influencing factors on local collaborative innovation between the two industries, but also evaluate the impact of local influencing factors on collaborative innovation between the two industries in surrounding areas.

Comparison of results under different models: This article constructs two models with geographical distance weight and economic distance weight. The results are shown in Model 1 and Model 2 of Table 6, respectively.

Robustness test: robustness test is carried out in two ways: replacing distance weight and adding control variable. See 5.4 robustness Test for details.

Comment 9（Reviewer #2：In the part of research status, the outline of the whole research is not clear enough, and more content of multi criteria decision model (method) needs to be added. The results of the application part of the model need to be rearranged, the readability is too poor, and the graphical results provided can’t make people see the differences under different scene settings.

Response 9: The research status section has been fully revised. According to the theme of this article, it has been divided into three parts: the definition of collaborative innovation, the influencing factors of collaborative innovation, and the research methods of collaborative innovation. The graphical results have been modified into a table.

---

## [Decision Letter · Decision Letter 1]

29 May 2023

Collaborative Innovation Evolution of Logistics and Manufacturing industry in China

PONE-D-23-00850R1

Dear Dr. Xiuling,

We’re pleased to inform you that your manuscript has been judged scientifically suitable for publication and will be formally accepted for publication once it meets all outstanding technical requirements.

Kind regards,

László Vasa, PhD

Academic Editor

PLOS ONE

Additional Editor Comments (optional):

Reviewers' comments:

Reviewer's Responses to Questions

**Comments to the Author**

1. If the authors have adequately addressed your comments raised in a previous round of review and you feel that this manuscript is now acceptable for publication, you may indicate that here to bypass the “Comments to the Author” section, enter your conflict of interest statement in the “Confidential to Editor” section, and submit your "Accept" recommendation.

Reviewer #2: All comments have been addressed

2. Is the manuscript technically sound, and do the data support the conclusions?

Reviewer #2: Yes

3. Has the statistical analysis been performed appropriately and rigorously? 

Reviewer #2: Yes

4. Have the authors made all data underlying the findings in their manuscript fully available?

Reviewer #2: Yes

5. Is the manuscript presented in an intelligible fashion and written in standard English?

Reviewer #2: Yes

6. Review Comments to the Author

Reviewer #2: All the reviewers' comments have been addressed carefully and sufficiently, the revisions are rational from my point of view, I think the current version of the paper can be accepted.

7. PLOS authors have the option to publish the peer review history of their article (what does this mean?). If published, this will include your full peer review and any attached files.

Reviewer #2: No

---

## [Editor Report · Acceptance letter]

7 Jun 2023

PONE-D-23-00850R1 

*Collaborative Innovation Evolution of the Logistics and Manufacturing Industry in China*

Dear Dr. Chen:

I'm pleased to inform you that your manuscript has been deemed suitable for publication in PLOS ONE. Congratulations! Your manuscript is now with our production department. 

Kind regards, 

on behalf of

Prof. Dr. László Vasa 

Academic Editor

PLOS ONE